# Clearance of Persistent SARS-CoV-2 RNA Detection in a NFκB-Deficient Patient in Association with the Ingestion of Human Breast Milk: A Case Report

**DOI:** 10.3390/v14051042

**Published:** 2022-05-13

**Authors:** Janine S. Sabino, Mariene R. Amorim, William M. de Souza, Lia F. Marega, Luciana S. Mofatto, Daniel A. Toledo-Teixeira, Julia Forato, Rodrigo G. Stabeli, Maria Laura Costa, Fernando R. Spilki, Ester C. Sabino, Nuno R. Faria, Bruno D. Benites, Marcelo Addas-Carvalho, Raquel S. B. Stucchi, Dewton M. Vasconcelos, Scott C. Weaver, Fabiana Granja, José Luiz Proenca-Modena, Maria Marluce dos S. Vilela

**Affiliations:** 1Laboratory of Pediatric Immunology, Center for Investigation in Pediatrics, Faculty of Medical Sciences, University of Campinas, Campinas 13083-887, Brazil; janine@unicamp.br (J.S.S.); l150152@dac.unicamp.br (L.F.M.); 2Laboratory of Emerging Viruses, Department of Genetics, Microbiology and Immunology, Institute of Biology, University of Campinas, Campinas 13083-862, Brazil; mariene.ramorim@gmail.com (M.R.A.); luciana.mofatto@gmail.com (L.S.M.); teixeiradatt@gmail.com (D.A.T.-T.); foratojulia@gmail.com (J.F.); fabi.granja@yahoo.com.br (F.G.); 3World Reference Center for Emerging Viruses, Department of Microbiology and Immunology, University of Texas Medical Branch, Galveston, TX 77555, USA; wmdesouz@utmb.edu (W.M.d.S.); sweaver@utmb.edu (S.C.W.); 4Oswaldo Cruz Foundation (Fiocruz-SP), Ribeirão Preto 14049-900, Brazil; rodrigo.stabeli@fiocruz.br; 5Department of Public Health Emergency, Preparedness and Disaster, PAHO/WHO, Brasilia 70312-970, Brazil; 6Department of Obstetrics and Gynecology, Faculty of Medical Sciences, University of Campinas, Campinas 13083-887, Brazil; mlaura@unicamp.br; 7One Health Laboratory, Feevale University, Novo Hamburgo 93510-235, Brazil; fernandors@feevale.br; 8Tropical Medicine Institute, Medical School, University of São Paulo, São Paulo 5403-907, Brazil; sabinoec@gmail.com; 9Department of Infectious and Parasitic Disease, Medical School, University of São Paulo, São Paulo 05403-000, Brazil; n.faria@imperial.ac.uk; 10Department of Zoology, University of Oxford, Oxford OX1 2JD, UK; 11MRC Centre for Global Infectious Disease Analysis, J-IDEA, Imperial College London, London SW7 2AZ, UK; 12Hematology and Transfusion Medicine Center, University of Campinas, Campinas 13083-878, Brazil; benites@unicamp.br (B.D.B.); maddas@unicamp.br (M.A.-C.); 13Division of Infectious Diseases, University of Campinas, Campinas 13083-887, Brazil; stucchi@unicamp.br; 14Laboratory of Investigation in Dermatology and Immunodeficiencies, Department of Dermatology, Medical School, University of São Paulo, São Paulo 05403-000, Brazil; dmvascon@usp.br; 15Institute for Human Infection and Immunity, University of Texas Medical Branch, Galveston, TX 77555, USA; 16Biodiversity Research Centre, Federal University of Roraima, Boa Vista 72000-000, Brazil; 17Experimental Medicine Research Cluster (EMRC), University of Campinas, Campinas 13083-862, Brazil; 18Hub of Global Health (HGH), University of Campinas, Campinas 13083-862, Brazil

**Keywords:** persistent SARS-CoV-2 infection, immunosuppression, NFκB-deficiency, breast milk, IgA

## Abstract

Currently, there are no evidence-based treatment options for long COVID-19, and it is known that SARS-CoV-2 can persist in part of the infected patients, especially those with immunosuppression. Since there is a robust secretion of SARS-CoV-2-specific highly-neutralizing IgA antibodies in breast milk, and because this immunoglobulin plays an essential role against respiratory virus infection in mucosa cells, being, in addition, more potent in neutralizing SARS-CoV-2 than IgG, here we report the clinical course of an NFκB-deficient patient chronically infected with the SARS-CoV-2 Gamma variant, who, after a non-full effective treatment with plasma infusion, received breast milk from a vaccinated mother by oral route as treatment for COVID-19. After such treatment, the symptoms improved, and the patient was systematically tested negative for SARS-CoV-2. Thus, we hypothesize that IgA and IgG secreted antibodies present in breast milk could be useful to treat persistent SARS-CoV-2 infection in immunodeficient patients.

## 1. Introduction

After more than two years since the beginning of the Coronavirus disease (COVID-19) pandemic, there have been several reports of patients with persistent symptoms, a condition known as “long COVID”. These symptoms include fatigue, breathlessness, hair loss, myalgia, and neurocognitive difficulties [1]. The mechanisms and risk factors associated with long COVID are not fully understood, although it is known that severe acute respiratory syndrome coronavirus 2 (SARS-CoV-2) can persist in part of the infected patients, especially those with immunosuppression [2]. However, the association between SARS-CoV-2 persistent infection and long COVID-19 has not been fully clarified. Immunoglobulin A (IgA) antibodies present on mucosal surfaces play a crucial role against respiratory viruses, such as Influenza, and have been associated with the efficacy of intranasal vaccines [3,4]. IgA-specific antibodies may also play a key role in the immune response to COVID-19. It has been shown that IgA dimers are more potent in neutralizing SARS-CoV-2 than immunoglobulin G (IgG) monomers [5] since they peak before IgG antibodies in serum samples, thus representing an early potent neutralizing agent against SARS-CoV-2 [6]. IgA and IgG antibodies against SARS-CoV-2 can be found in the breast milk of vaccinated women or women previously infected with COVID-19 and have shown neutralizing capacity in vitro [7].

This study reports the effect of plasma infusion followed by breast milk ingestion as a form of treatment for persistent SARS-CoV-2 infection in a patient with a common variable immunodeficiency (CVID) phenotype, with polymorphism in the NFκB1 gene [Q202X (c.604 C > T)], which is usually associated with immunosuppression and severe B-cell and antibody production deficiency [8]. The patient had a history of recurrent infections, hypogammaglobulinemia, low IgG levels, absence of IgA, neutropenia, and thrombocytopenia since age 7 and was submitted to splenectomy with improvement in neutropenia and thrombocytopenia, as described elsewhere [8]. Since there are no evidence-based treatment options for patients with long-lasting symptoms, nor for persistent SARS-CoV-2 infection [1,2], we hypothesized that IgA and IgG-secreted antibodies present in breast milk could be useful to treat persistent SARS-CoV-2 infection in immunodeficient patients.

## 2. Materials and Methods

### 2.1. Ethics

All procedures followed the ethical standards of the responsible committee on human experimentation and were approved by the ethics committee of the University of Campinas (Campinas, Brazil; approval numbers CAAE: 53160821.0.0000.5404; CAAE: 17187819.4.0000.5404; and CAAE: 30580020.1.0000.0008). Patient and participant data were anonymized, and written informed consent was obtained.

### 2.2. Patient

The patient in this study was a 32-year-old female diagnosed with a single nucleotide polymorphism in the NFκB1 gene (NM_001165412: exon 8:c.C604T:p.Q202X, NFKB1:NM_003998:exon8:c. C607T:p.Q203X), which is known to affect the Rel homology domain (RHD) of the NFκB1 gene, responsible for mediating DNA-binding, nuclear translocation, and dimerization [8]. Her clinical phenotype is characterized for lymphoproliferation (splenomegaly), autoimmunity (cytopenia), absent serum IgA and low IgM and IgG levels, reduced peripheral CD3+ and CD4+ cells, and an absence of CD19+ cells. Impaired canonical NFκB signaling has been observed in NFκB1-haploinsufficient patients, suggesting that defective protein synthesis might occur. As a consequence, the immune and inflammatory responses, as well as cell survival, are impaired [8].

### 2.3. COVID-19 Convalescent Plasma

Convalescent plasma units were obtained from two donors by apheresis using the Amicus™ automated blood cell separator (Fresenius Kabi AG, Bad Homburg vor der Höhe, Germany). Donation criteria included confirmation of previous SARS-CoV-2 infection through positive reverse transcription real-time polymerase chain reaction (RT-PCR) test results, absence of symptoms for at least 28 days, and eligibility for other clinical and laboratory blood donation criteria, in accordance with national legislation. The evaluation of SARS-CoV-2-specific neutralizing antibody titers was performed by observing the absence of cytopathic effect in cultures of Vero E6 cells incubated with a serum–virus mixture [9]. After three days of incubation, the Vero cells were inspected using an inverted optical microscope; the highest serum dilution that protected more than 80% of cells from cytopathic effect was taken as the neutralization titer. These procedures were carried out within a clinical trial approved by the Brazilian Commission on Ethics in Research (approval number CAAE: 30580020.1.0000.0008). Both donors were blood type O+, with neutralizing antibody titers of 1:640. The SARS-CoV-2 genomes of the blood donors were not sequenced, but it is likely that both of them were infected with the VOC Gamma SARS-CoV-2 lineage since this variant and its sub-lineages were the most prevalent in Brazil (corresponding to 89.81% of the deposited genomes on GSAID) during March and April 2021, a period in which both donors had COVID-19 (https://www.gisaid.org/, accessed on 26 April 2022) [10].

### 2.4. Human Breast Milk

Human breast milk from a fully COVID-19-vaccinated mother was obtained from a maternal milk bank in São Paulo, Brazil, after ethical approval. The criteria for the selection of the milk donor were the absence of clinical symptoms, previous SARS-CoV-2-complete vaccination for at least 15 days, and negative test results for human immunodeficiency virus (HIV) and hepatitis B and C virus (HBV and HCV). The milk administered in this study was previously pasteurized. The procedure for milk ingestion includes intaking a 30 mL dose of milk every 4 h for 14 days, totaling a 180 mL volume per day. The proband should retain the milk in their mouth for 3 min before swallowing and repeat the RT-PCR for SARS-CoV-2 after 14 days of treatment. In this study, breast milk from only one donor was used. The milk donor appointed herein received two doses of the BNT162b2 vaccine (Pfizer–BioNTech, Kalamazoo, MI, USA). The second dose was applied on 5 June 2021. The donor reported not having had COVID-19.

### 2.5. RNA Extraction and RT-qPCR

Viral RNA was extracted from nasopharyngeal swab samples using the Quick-DNA/RNA viral kit (Catalog number: D7021, Zymo Research, Irvine, CA, USA) according to the manufacturer’s instructions. RT-PCR for SARS-CoV-2 diagnosis was performed in the Clinical Pathology Laboratory (LPC) of the University of Campinas Hospital with the GeneFinder COVID-19 Plus RealAmp Kit (Catalog number: IFMR-45, OSANG Healthcare, Anyang, South Korea) [11], following the manufacturer’s instructions. For viral load quantification, RNA was tested by reverse transcription real-time quantitative polymerase chain reaction (RT-qPCR) targeting the envelope gene [12].

### 2.6. SARS-CoV-2 Genomic Sequencing and Phylogenetic and Genomic Analyses

Three RNA samples were submitted to SARS-CoV-2 genome sequencing, carried out using the ARTIC network SARS-CoV-2 V3 primer scheme (https://artic.network/ncov-2019, accessed on 26 April 2022) protocol with the MinION platform (Oxford Nanopore Technologies, Oxford, UK) [13,14]. FAST5 files containing the raw signal data were basecalled, demultiplexed, and trimmed using Guppy software, version 4.4.1 (Oxford Nanopore Technologies, Oxford, UK). The reads were aligned against the reference genome Wuhan-Hu-1 (GenBank accession No. MN908947.3) using the minimap2 software, version 2.17.r941 (Cambridge, MA, USA) [15], and converted into a sorted BAM file using the SAMtools software (Cambridge, MA, USA) [16]. Length filtering (minimum and maximum sizes of 400 bp and 700 bp, respectively), quality testing, primer trimming, variant calling, and consensus sequencing were performed for each barcode using guppyplex and the ARTIC nanopolish pipeline (https://artic.network/ncov-2019/ncov2019-bioinformatics-sop.html, accessed on 26 April 2022). Genome regions with a depth of <20-fold were represented with N characters. Subsequently, the genomes were uploaded to the CoV-GLUE online resource (http://cov-glue.cvr.gla.ac.uk, accessed on 26 April 2022) and Pangolin web application (https://pangolin.cog-uk.io/, accessed on 26 April 2022) [17] for mutation determination and lineage classification. Finally, we used the open-source interactive tool NextClade(v0.4.0) (Seattle, WA, USA) [18] to assign the sequences to clades.

### 2.7. Flow Cytometry Analysis

Lymphocyte subset analysis was performed by flow cytometry, with a “stain-lyse-wash” protocol, using the following monoclonal antibody combination, on 19th May: CD4 FITC (Catalog #555346, BD Biosciences, Franklin Lakes, NJ, USA), TCRab PE (Catalog #555548, BD Biosciences), CD3 PerCP (Catalog # 347344, BD Biosciences), CD56 APC (Catalog #17-0566-42, eBioscience, San Diego, CA, USA), CD16 APC (Catalog #561248, BD Biosciences), CD8 APC-eFluor780 (Catalog #47-0088-42, Invitrogen), CD19 eFluor 450 (Catalog # 48-0198-42, Invitrogen, Waltham, MA, USA), TCRgd PE Cy7 (Catalog #655410, BD Biosciences), and CD45 eFluor506 (Catalog #69-0459-42, Invitrogen). On 12th July, the following monoclonal antibody combination was used: CD3FITC/CD8PE/CD45PERCP/CD4APC (Catalog #340499, BD Biosciences), CD3FITC/CD16+CD56PE/CD45PERCP/CD19APC (Catalog #340500, BD Biosciences), CD4FITC (Catalog #555346, BD Biosciences), CD45ROPE (Catalog #555493, BD Biosciences), CD45RAAPC (Catalog #550855, BD Biosciences), and FACS Lysing Solution (Catalog #349202, BD Biosciences). A total of 100,000 cells were acquired in a FACSCanto II flow cytometer (BD Biosciences, Franklin Lakes, NJ, USA). The analysis was conducted using the FACsDiva software (BD Biosciences, Franklin Lakes, NJ, USA). The percentage of lymphocyte subsets was calculated in the gate CD45high/SSClow. B lymphocytes were CD19+, T lymphocytes were CD3+, and NK were CD3-/CD56+ cells.

## 3. Case Report

On 11 March 2021, the patient reported the onset of COVID-19 symptoms, including fever (38–38.5 °C), headache, asthenia, anorexia, diarrhea, sore throat, and cough, which lasted approximately two weeks. On 15 March 2021, SARS-CoV-2 RNA was detected in a nasopharyngeal sample by RT-qPCR. One month later, the patient remained symptomatic, and nasopharyngeal swab samples presented a viral load of 1.99 × 10^7^ RNA copies per mL. Then, based on a near-complete sequence with 87.4% genome coverage (20× depth coverage), the SARS-CoV-2 was classified as the Gamma variant (P.1 PANGO lineage) (Appendix A). Although the patient remained febrile for 75 days (varying from 37.8 °C to 38.5 °C according to daily measurement before bed with an axillary thermometer), with slight weight loss (~12.5% in 120 days), chest tomography, electrocardiogram, and oxygen saturation were normal. Other symptoms, such as headache, anorexia, and cough, were also recurrently present during these 75 days in the patient. The diagnosis of prolonged infection was based on the detection of SARS-CoV-2 RNA on 15 May (3.06 × 10^7^ RNA copies/mL) and 7 June (1.35 × 10^8^ RNA copies/mL), nearly 3 months after the initial diagnosis, by RT-PCR and confirmed by genome whole-sequencing (Figure 1). We performed a maximum-likelihood phylogenetic analysis using NextStrain (Auspice tool) [8] and confirmed all Gamma lineage signatures (Appendix A) using the CoV-GLUE online resource. We did not find nucleotide substitutions in the three SARS-CoV-2 genomes in the patient, thus excluding potential re-infection.

The clinical and laboratory parameters of the patient did not meet the criteria for hospitalization; therefore, she remained self-isolated at home, under medical supervision. Our patient’s clinical course of SARS-CoV-2 infection can be defined as dysregulated but without hyperinflammation (124 days).

On 19 May 2021, the patient was infused with a total of 600 mL of plasma from two COVID-19 convalescent patients by the intravenous route (IgG titers: 1:640) for two consecutive days. Three days after COVID-19 convalescent plasma infusion, she reported feeling more active and stronger, without persistent fever. However, SARS-CoV-2 RNA was still detectable by RT-qPCR in oropharyngeal swab samples collected on 7 June 2021, after the COVID-19 convalescent plasma infusion (Figure 1). Therefore, the patient remained infected and was at risk of infecting other people and even progressing to more severe COVID-19 since she still felt weak and had not yet recovered her appetite.

Since IgA dimers are more potent in neutralizing SARS-CoV-2 in vitro than IgG [5], a treatment providing IgA specific for SARS-CoV-2 for the patient was suggested, via ingestion of human mature breast milk from COVID-19-vaccinated women at ~0.1 g/dL. On 7 July 2021, the patient started treatment with the intake of pasteurized breast milk from a BNT162b2-vaccinated (Pfizer–BioNTech) mother every 4 h (180 mL per day) for two weeks. The procedure included retaining the milk in her mouth for at least 3 min, followed by swallowing. 

It was noticed that one week after convalescent plasma infusion, the absolute numbers of neutrophils (4170/mm^3^), lymphocytes (1320/mm^3^), and monocytes (1170/mm^3^) practically doubled (7880, 2400, and 2400, respectively/mm^3^), indicating the recovery of the hematopoietic system. However, one week later, the number of lymphocytes was reduced to half (1320/mm^3^), returning to a value near the upper limit of normality (3730/mm^3^) after ingesting the vaccinated mother’s milk (Table 1). This fluctuation observed in the number of neutrophils, lymphocytes, and monocytes probably corresponded to the clinical status of symptomatic infection followed by the impact of the therapeutic interventions.

In addition, before COVID-19 convalescent plasma infusion, the absolute number of CD3+ and CD4+ cells was reduced by >50% compared to the usual values for this patient, but after plasma infusion and breast milk ingestion treatment, the amounts of CD3+ and CD4+ were normalized. However, the CD4+/CD8+ ratio and the absolute number of CD19+ remained unchanged during the clinical course. After both interventions, the marked reduction in the elevated serum ferritin and C-reactive protein levels were indicative of strong neutralizing activity, no doubt suggesting a protective effect against infection in this immunodeficient patient (Appendix A).

On 12 July 2021, the patient was fully recovered, and oropharyngeal swab samples tested negative by RT-qPCR 16 days after the end of treatment. In addition, the patient was still negative by RT-qPCR and IgM non-reagent for SARS-CoV-2 on 24 November 2021. Thus, albeit rare, this case supports the potential therapeutic use of specific secretory antibodies (IgA and IgG) from human breast milk to treat COVID-19 in CVID patients. We suspect that the IgA antibodies present in the breast milk of vaccinated women can neutralize SARS-CoV-2 maintained in the tonsils and oropharyngeal and gastrointestinal tracts, which may be important sites of viral persistence in immunosuppressed individuals [19,20,21,22]. This case report provides evidence that suggests the milk may have neutralized the virus and/or led to the subsequent clearance of infection and highlights the importance of further studies in this field to help immunodeficient patients with long COVID.

## 4. Discussion

Impairments in innate or adaptive immune response are key points in SARS-CoV-2 persistent infection. In fact, results obtained in a short series of case reports of COVID-19 in immunocompromised patients suggest that SARS-CoV-2 infections may be maintained in these individuals even with high titers of NAb post-vaccination [22]. Thus, immunocompromised patients are at risk of developing persistent infection [23]. In addition, there is no evidence-based treatment available for SARS-CoV-2 persistent infection or long COVID-19 [1,2]. This study reports a case of persistent infection in a patient with the CVID phenotype, with a polymorphism in the NFκB1 gene, which affects B cell development and the production of IgA and IgG. 

IgA antibodies play an important role against respiratory viral infections and are early secreted in the course of SARS-CoV-2 infection, followed by IgM and IgG antibodies [4,5,6]. High levels of specific anti-RBD IgA are detected in the serum, saliva, and bronchoalveolar lavage (BAL) of COVID-19 patients, suggesting an important role against infection of the mucosa [6]. Breast milk from vaccinated mothers also shows a rapid increase in specific anti-SARS-CoV-2 IgA antibodies, followed by IgG, 2 weeks after the first vaccine dose, which increases at 4 weeks and after the second dose of the Pfizer-BioNTech vaccine [24]. The patient in this study was diagnosed with persistent infection, with high viral load, caused by the Gamma variant (PANGO Lineage P.1). She received convalescent plasma intravenously, which helped in symptom relief. However, the infection and symptoms were completely resolved only after treatment with the vaccinated mother’s breast milk via the oral route. Thus, we believe that the IgG and, especially, IgA antibodies present in the milk act by inactivating viruses present in the mucosa of the gastrointestinal tract. In addition, although breast milk antibodies cannot pass into serum in humans [25], they can confer systemic protection against different pathogens, as observed for HIV and respiratory viruses [26]. Today, we know that breast milk antibodies are essential for selecting the gut microbiome, which is critical for immune defense and immune education [25]. For example, short-chain fatty acids produced by commensal bacteria in the gut, which can be potentially modulated by breast milk antibodies, are important for the regulation of type I interferon, which acts in the control of Respiratory Syncytial Virus (RSV) infection [27].

It is important to highlight that hyperinflammation was not observed in the course of infection in our patient, which lasted 124 days. It is well known that host factors associated with innate immune response against cellular infection, such as the imbalanced production of inflammatory cytokines, can lead to acute respiratory distress syndrome in COVID-19 patients and that inflammatory imbalance is associated with systemic symptoms [28]. In addition, it was noted that SARS-CoV-2 infected cells showed a prevalent NFκB transcriptional signature, high expression of proinflammatory cytokines, and a viral replication dependency of NFκB transcription factor p65 or p50, suggesting that the inflammatory profile observed in infected cells depends on host factors of the NFκB gene family. Thus, the absence of severe symptoms could be associated with the polymorphism observed in the patient of this case report [Q202X (c.604 C > T)] [8].

Recent studies have shown that SARS-CoV-2 messenger RNA (mRNA) vaccines, such as BNT162b2 (Pfizer-BioNTech) and mRNA-1273 (Moderna), are able to elicit high levels of neutralizing antibodies [29], usually higher than those observed after other vaccines [30]. Moreover, the levels of anti-spike antibodies are maintained at high titers for several months after mRNA vaccination, mainly in previously infected individuals who received at least two doses of mRNA vaccine, or three independent exposures to the spike antigen [31], in an event associated with the formation of a persistent germinal center B cell response [32]. Similarly, the anti-SARS-CoV-2 IgA and IgG antibody titers in the milk of vaccinated mothers did not decrease after 60 days of complete vaccination with mRNA vaccines [33].

There is a degree of cross-reactive natural immunity induced by infection with different SARS-CoV-2 variants or vaccination that is increased in infected individuals with a history of full vaccination [34]. However, new VOC variants, such as B.1.1.529 (omicron), BA.1, and BA.2, have demonstrated the ability to escape from neutralizing antibodies induced after natural infection or vaccination. The neutralizing titers detected in the serum of vaccinated individuals who received two doses of the BNT162b2 vaccine are usually lower for the omicron variant than for original SARS-CoV-2 lineages. Interestingly, this difference was six times lower in individuals who received three doses of the vaccine or who had a history of complete vaccination and infection [35]. Thus, further studies investigating the efficacy of milk ingestion from donors who have different histories of vaccination or infection are warranted.

Finally, persistently infected immunosuppressed patients have been linked to virus evolution and the possible emergence of new variants [36,37,38]. In this case report, we sequenced three samples collected during infection persistence and did not observe any new mutations or single nucleotide polymorphisms (SNPs) in the Gamma variant genome detected in this patient. 

Limitations of this study include the rarity of the reported immunodeficiency, which limits generalizations to other types of immunosuppression or hypogammaglobulinemia; the absence of testing for neutralizing antibody titers and the number of different classes of antibodies against specific regions of the virus in the breast milk of the donor; and the fact that the patient was not tested for SARS-CoV-2 by RT-qPCR immediately before breast milk ingestion. Therefore, it is possible that other factors aside from breast milk ingestion may have contributed to virus clearance in the persistently infected woman.

## 5. Conclusions

Breast milk from COVID-19-vaccinated mothers or specific anti-SARS-CoV-2 IgA antibodies could be beneficial in the treatment of persistent infection in immunocompromised patients.

## Figures and Tables

**Figure 1 viruses-14-01042-f001:**
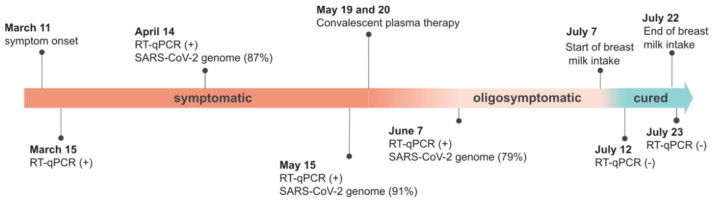
Timeline of COVID-19 in the NFκB-deficient patient. (+), positive; (−), negative. (%) represents the percentage coverage of the sequenced SARS-CoV-2 genome.

**Table 1 viruses-14-01042-t001:** Patient’s hemogram test results.

Parameters	19 May 2021	27 May 2021	7 June 2021	22 July 2021	Reference Values
WBC (×10^3^/μL)	6.78	13.09	10.96	9.57	4–10
RBC (×10^6^/μL)	4.14	4.09	4.55	5.40	4.2–5.4
Hb (g/dl)	12.10	12.40	13.80	15.30	12–16
Ht (%)	37.10	37.90	42.90	46.20	37–47
MCH (fl)	89.60	92.70	94.30	85.50	80–99
MCHC (pg)	29.20	30.30	30.30	28.30	27–32
CHCM (g/dl)	32.60	32.70	32.20	34.20	32–37
RDW (%)	13.50	15.40	16.80	15.20	10–15
PLT (×10^3^/μL)	260	292	206	299	150–400
MPV (fl)	12.20	12.40	12.20	10.80	6–10
Neutrophil/mm^3^	4170	7880	9070	5010	2000–8000
Lymphocyte/mm^3^	1320	2400	1320	3730	1000–4000
Monocyte/mm^3^	1170	2400	550	680	200–800
Eosinophil/mm^3^	20	130	0	80	0–450
Basophil/mm^3^	20	0	0	7	0–200

Legend: WBC, white blood cells. RBC, red blood cells. Hb, hemoglobin. Ht, hematocrit. MCH, mean corpuscular hemoglobin. MCHC, mean corpuscular hemoglobin concentration. RDW, red blood cell distribution width. PLT, platelet count. MVP, mean platelet volume.

## Data Availability

The SARS-CoV-2 sequences obtained in this case report have been uploaded to the GISAID Initiative (www.gisaid.org, accessed on 26 April 2022) under the accession numbers EPI_ISL_6513196 (14 April 2021), EPI_ISL_6513480 (15 May 2021), and EPI_ISL_6513725 (7 June 2021).

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
