# Peer review of "Clearance of Persistent SARS-CoV-2 RNA Detection in a NFκB-Deficient Patient in Association with the Ingestion of Human Breast Milk: A Case Report"

_viruses, 2022, doi:10.3390/v14051042_

Round 1

Reviewer 1 Report

Thank you for asking me to review this manuscript. A Case Report is described in which a patient with an immunodeficiency with an NFkB1 polymorphism, absent serum IgA, low IgM and IgG levels has persistent Gamma variant SARS-CoV-2 infection which only resolves after ingestion of human breast milk, but not convalescent plasma infusion. The virology is well worked up with good demonstration of persistent infection with high viral loads.

The clinical features are described but it would be useful to know in section 3 a bit more detail behind 'the patient was febrile for 75 days' - was this subjective fever or objective fevers with actual temperature readings with a range that could be included in the text? During this follow up period it would be useful to comment specifically about the presence or absence of any sore throat, nasal or gastrointestinal symptoms. This is relevant to comments later on. 

For an international readership it would be useful to know more about the plasma infusion. This was derived from two convalescent patients. Is the variant that they were infected with known, was it Gamma, what other possible variants were circulating at the time in Brazil? It would be worth commenting on heterologous antibody neutralisation between different variants, especially as the poor cross-protection with Omicron is in people's minds at the moment.

The breast milk was derived post Pfizer vaccination and the implication is that the stimulated antibody was cross protective. By way of explanation could the authors expand on the extent of IgG and IgA mucosal absorption after ingestion. This will give context to the comment that it is proposed that mucosally retained IgA is thought to be the main agent for clearance of infection. Holding breast milk in the mouth and then swallowing would not have exposed the nasal epithelium or lower respiratory tract. 

In extrapolating from this case the authors should expand more on the point that this may only be relevant to patients who are a/hypogammaglobulinaemic. 

Author Response

Reviewer 1

Query 1. Thank you for asking me to review this manuscript. A Case Report is described in which a patient with an immunodeficiency with an NFkB1 polymorphism, absent serum IgA, low IgM and IgG levels has persistent Gamma variant SARS-CoV-2 infection which only resolves after ingestion of human breast milk, but not convalescent plasma infusion. The virology is well worked up with good demonstration of persistent infection with high viral loads.

R1. We thank for your valuable feedback. We worked hard in an attempt to characterize this persistent SARS-CoV-2 infection. We believe that the fact that we did not find any mutations in this virus in samples collected at different moments is a good indication that the patient was persistently infected and not recurrently infected by different variants of SARS-CoV-2. 

Query 2. The clinical features are described but it would be useful to know in section 3 a bit more detail behind 'the patient was febrile for 75 days' – was this subjective fever or objective fevers with actual temperature readings with a range that could be included in the text? During this follow up period it would be useful to comment specifically about the presence or absence of any sore throat, nasal or gastrointestinal symptoms. This is relevant to comments later on.

R2. Thank you for your observations. We agree with the reviewer that more details on the maintenance of fever and other clinical conditions would help to better understand the patient’s clinical course. The fever in this proband was maintained for 75 consecutive days, measured every day before bed using a digital axillary thermometer. In fact, the patient received this orientation to measure temperature daily or if worsening of clinical symptoms occurred. Temperatures were recorded in a notebook and ranged from 37.8°C to 38.5°C. After the plasma infusion, the persistent fever ceased, but other unspecific symptoms were maintained until before the breast milk ingestion. We have provided more details regarding fever and other symptoms in the first paragraph of Section 03 of the reviewed manuscript (highlighted in yellow).

Query 3. For an international readership it would be useful to know more about the plasma infusion. This was derived from two convalescent patients. Is the variant that they were infected with known, was it Gamma, what other possible variants we recirculating at the time in Brazil? It would be worth commenting on heterologous antibody neutralization between different variants, especially as the poor cross-protection with Omicron is in people's minds at the moment.

R3. Thank you, this issue was very well pointed out by the reviewer. A new sentence was added in Section 2.3 (highlighted in yellow) in order to provide more details regarding the process of plasma infusion. Unfortunately, the SARS-CoV-2 genome from the plasma donors has not yet been sequenced. From the epidemiological data, we can infer that both donors were infected with the Gamma variant of SARS-CoV-2 since this VOC was the predominant lineage of the virus during April 2021, when both donors developed symptoms of COVID-19. In addition, we added a sentence about antibody cross-protection, mainly regarding the Omicron variant, in the Discussion. Please see the second-to-last paragraph highlighted in yellow in the Discussion section.

Query 4. The breast milk was derived post Pfizer vaccination and the implication is that the stimulated antibody was cross protective. By way of explanation could the authors expand on the extent of IgG and IgA mucosal absorption after ingestion. This will give context to the comment that it is proposed that mucosally retained IgA is thought to be the main agent for clearance of infection. Holding breast milk in the mouth and then swallowing would not have exposed the nasal epithelium or lower respiratory tract.

R4. Patients with an absence of serum IgA also do not produce secretory IgA. Maternal milk with normal serum IgA has high concentrations of secretory IgA, whose function is to neutralize the antigen and clear it through saliva and feces. Because of the anti-inflammatory properties of secretory IgA, this therapeutic intervention acted as a replacement of secretory IgA for the patient, further benefiting her as it is a specific SARS-CoV-2 IgA. Thus, we believe that the ingestion of milk every three hours probably enabled the proband to maintain a regular concentration of this secretory IgA and the clearance of SARS-CoV-2 from its gastrointestinal tract. However, we agree with the reviewer. We cannot exclude the possibility that systemic effects of these antibodies may have contributed, even indirectly, to the clinical improvement and viral clearance of the patient. Thus, we added a sentence regarding the systemic effect of breast milk antibodies modulated by intestinal microbiota in the Discussion section. Please see the last sentence added in the second paragraph of the Discussion, in Section 4 (highlighted in yellow).

Query 5. In extrapolating from this case, the authors should expand more on the point that this may only be relevant to patients who are hypogammaglobulinemic.

R5. Thank you for your comment. We have added a sentence regarding this issue in the Discussion section. Please see the highlighted sentence in the last paragraph of the Discussion.

Reviewer 2 Report

The authors are presenting a very interesting case of a 32-year-old female patient diagnosed with a single nucleotide polymorphism in the NFκB1 gene and with persistent SARS-CoV-2 infection.

The authors thought that IgA and IgG secreted antibodies present in the breast milk could be useful to treat persistent SARS-CoV-2 infection in immunodeficient patients.

The patient was followed- up for a period of 124 days 7 by q RTPCR for SARS-CoV-2 genome assessing tests.

SARS-CoV-2 analysed by genomic sequencing, phylogenetic and genomic testing and lymphocyte subset analysis was performed by flow cytometry. The hemogram, ferritin, CPR of the patients were also monitored in this follow up period.

In this case presentation, the authors considered that breast milk from COVID-19 vaccinated mothers or specific anti-SARS-CoV-2 IgA antibodies could be beneficial to the treatment of persistent infection in immunocompromised patients.

If the authors and editor consider adequate, I kindly  suggest  citing this paper - DOI: 10.3390/pathogens11030286, with a close  research subject like this case presentation.

I would like to congratulate the authors for this comprehensive analysis and follow-up of this special case of immunodeficient patient.

Author Response

Query 1. The authors are presenting a very interesting case of a 32-year-old female patient diagnosed with a single nucleotide polymorphism in the NFκB1 gene and with persistent SARS-CoV-2 infection. The authors thought that IgA- and IgG-secreted antibodies present in the breast milk could be useful to treat persistent SARS-CoV-2 infection in immunodeficient patients. The patient was followed-up for a period of 124 days by qRT-PCR for SARS-CoV-2 genome-assessing tests. SARS-CoV-2 analyzed by genomic sequencing, phylogenetic and genomic testing, and lymphocyte subset analysis was performed by flowcytometry. The hemogram, ferritin, PCR of the patients were also monitored in this follow-up period. In this case presentation, the authors considered that breast milk from COVID-19-vaccinated mothers or specific anti-SARS-CoV-2 IgA antibodies could be beneficial to the treatment of persistent infection in immunocompromised patients.

R1. Thank you for your valuable feedback.

Query 2. If the authors and editor considered equate, I kindly suggest citing this paper - DOI: 10.3390/pathogens11030286, with a close research subject like this case presentation?

R2. Thank you for your suggestion. We have added a sentence regarding this paper in the Discussion section of the manuscript. Please see the last sentence of the fourth paragraph of the Discussion section. This paper was properly cited.

Query 3. I would like to congratulate the authors for this comprehensive analysis and follow-up of this special case of immunodeficient patient.

R3. We thank the reviewer for such a kind comment.

Reviewer 3 Report

Sabino and coauthors present an interesting case report of a patient presenting with long COVID symptoms that despite receiving convalescent plasma treatment remained RT-qPCR positive for SARS-CoV-2 RNA. Approximately 1 month later the patient received pasteurized donor breast milk from an individual that received a full course of a COVID-19 vaccine; and upon RT-qPCR testing a week later returned a negative result for SARS-CoV-2 RNA. The authors conclude that the breast milk helped to clear the SARS-CoV-2 infection. While this observation may be true, the fact that no RT-qPCR testing of the patient occurred prior to consumption of the breast milk makes it difficult to attribute the clearance of SARS-CoV-2 viral RNA and/or virus from the patient. It is also equally likely that the patient had cleared the viral RNA in the preceding month since their last RT-qPCR test. This limitation and alternative hypothesis is not discussed at all and should be further addressed. Presenting RT-qPCR data of a specimen collected just prior to the patient consuming the breast milk would dramatically strengthen the conclusion of the authors and alleviate concerns about alternate pathways for viral clearance.

Author Response

Query 1. Sabino and coauthors present an interesting case report of a patient presenting with long COVID symptoms that despite receiving convalescent plasma treatment remained RT-qPCR positive for SARS-CoV-2 RNA. Approximately 1 month later the patient received pasteurized donor breast milk from an individual that received a full course of a COVID-19 vaccine; and upon RT-qPCR testing a week later returned a negative result for SARS-CoV-2 RNA. The authors conclude that the breast milk helped clear the SARS-CoV-2 infection. While this observation may be true, the fact that no RT-qPCR testing of the patient occurred prior to consumption of the breast milk makes it difficult to attribute the clearance of SARS-CoV-2 viral RNA and/or virus from the patient. It is also equally likely that the patient had cleared the viral RNA in the preceding months since their last RT-qPCR test. This limitation and alternative hypothesis is not discussed at all and should be further addressed. Presenting RT-qPCR data of a specimen collected just prior to the patient consuming the breast milk would dramatically strengthen the conclusion of the authors and alleviate concerns about alternate pathways for viral clearance.

R1. Thank you for your valuable feedback. The reviewer is correct; this was the main limitation of this study. Nasopharyngeal swabs were not collected immediately prior to breast milk ingestion. Thus, unfortunately, we were unable to conduct the RT-qPCR testing of the patient before breast milk consumption. Hence, there is a possibility that the SARS-CoV-2 clearance was induced by plasma infusion or some other reason. However, we would like to mention that, although the patient improved after treatment with plasma infusion, the symptom only completely ceased after the ingestion of breast milk. In order to clarify these points, we included a sentence with the limitations of the study, highlighting the lack of this testing before breast milk ingestion. Please see the last sentence of the Discussion section (highlighted in yellow). In addition, we modified the Figure of the paper, changing asymptomatic to oligosymptomatic.

Round 2

Reviewer 3 Report

I appreciate the authors revisions and believe that they have improved the manuscript. However, I feel that based on the findings in the manuscript that the article title is misleading as it implies causation (Secretory antibodies from human breast milk eliminate SARS-CoV-2 persistent infection) and should be corrected to accurately reflect the findings of the study. As breast milk antibodies were not measured in this study and neutralizing or viral viability tests were not performed, the title should be corrected to reflect observational nature of the case report and the association that was identified. Something along the line of, e.g., Clearance of persistent SARS-CoV-2 RNA in association with consumption of donor human milk in a NFκB-deficient patient: a case report.

In addition, causative language is also used in the final sentence of the results and should be corrected: "This case report provides evidence of the effective neutralizing response of IgA and IgG antibodies present in breast milk and highlights the importance of further studies in this field to help immunodeficient patients with long COVID." This case reports provides evidence that suggests the milk may have neutralized the virus and/or led to the subsequent clearance. 

There appears to be a type in the final paragraph of the discussion: "the absence of neutralizing antibody titers and number of different..." should be "the absence of testing for neutralizing antibody titers and number of different..."

Author Response

Thank you for your valuable feedback. You are absolutely right! We have accepted all suggestions. The paper is now entitled: Clearance of persistent SARS-CoV-2 RNA detection in a NFκB-deficient patient in association with the ingestion of human breast milk: a case report. In addition, the last sentence of results and discussion were properly edited.  Please see these sentences highlighted in yellow.
